# Effect of Chronically Suppressed Plasma Angiotensin II on Regulation of the CYP4A/20-HETE Pathway in the Dahl Salt-Sensitive Rat

**DOI:** 10.3390/antiox12040783

**Published:** 2023-03-23

**Authors:** Kathleen Lukaszewicz, John R. Falck, Julian Lombard

**Affiliations:** 1Department of Physical Therapy, Marquette University, Milwaukee, WI 53233, USA; 2Department of Biochemistry, University of Texas Southwestern Medical School, Dallas, TX 75390, USA; 3Department of Physiology, Medical College of Wisconsin, Milwaukee, WI 53226, USA

**Keywords:** Dahl salt-sensitive rat, reactive oxygen species, cytochrome P450-4A (CYP4A), 20-HETE, high-salt diet, cerebral circulation, cerebral arteries

## Abstract

In Dahl salt-sensitive (SS) rats, impaired vascular relaxation can be restored by: (**1**) minipump infusion of a low (sub-pressor) dose of angiotensin II (ANG II) to restore physiological levels of plasma ANG II, (**2**) inhibition of 20-HETE production, and (**3**) introgression of a normally functioning renin allele from the Brown Norway rat (SS-13^BN^ consomic rat). Unlike SS rats, SS-13^BN^ rats have normal levels of ANG II on a normal-salt diet and suppressed ANG II on a high-salt (HS) diet. This study tested whether chronically low ANG II levels in SS rats upregulate cytochrome P450-4A (CYP4A) increasing the production of the vasoconstrictor 20-HETE. Although salt-induced suppression of ANG II levels increased reactive oxygen species (ROS) in basilar arteries from SS-13^BN^ rats in previous studies, this study showed no change in vascular 20-HETE levels in response to ANGII suppression. CYP4A inhibition significantly reduced vascular ROS levels and restored endothelium-dependent relaxation in response to acetylcholine in the middle cerebral artery (MCA) of SS rats and HS-fed SS-13^BN^ rats. These data demonstrate that both the renin–angiotensin system and the CYP4A/20-HETE pathway play a direct role in the vascular dysfunction of the Dahl SS rat but are independent of each other, even though they may both contribute to vascular dysfunction through ROS production.

## 1. Introduction

The SS rat has a low-renin salt-sensitive form of hypertension—the type of hypertension most prevalent in the African American population [1,2]. Human subjects [3,4] and animal models [5,6] with this form of hypertension have a reduced activity of the renin–angiotensin system leading to chronic suppression of plasma angiotensin II (ANG II) levels. Previous studies in our laboratory have shown that chronic exposure to low levels of circulating angiotensin II (ANG II) during elevated dietary salt intake leads to impaired vascular relaxation [7,8,9,10]. This is accompanied by reduced nitric oxide (NO) levels [10,11,12], elevated superoxide (O_2_^●−^) levels [10,11,12], and impaired endothelial Ca^2+^ signaling [12]. ANG II suppression leads to vascular oxidative stress by compromising the antioxidant defense mechanisms by suppressing Cu/Zn SOD expression [13,14,15]. All of these defects can be ameliorated by chronic intravenous (i.v.) infusion of a low (sub-pressor) dose of ANG II that restores normal circulating ANG II levels [7,8,9,10].

In a similar fashion, SS rats have an impaired ability to regulate their plasma renin activity (PRA) [16] and exhibit severe endothelial dysfunction and impaired vascular relaxation, even when they are normotensive and maintained on a normal-salt (NS; 0.4% NaCl) diet [17,18]. Impaired vascular relaxation in SS rats fed a NS diet can be restored not only by chronic i.v. infusion of a sub-pressor dose of ANG II to restore normal plasma levels of ANG II [7,8,19], but also by introgressing chromosome 13 from the Brown Norway (BN) rat carrying a normally functioning renin allele into the SS genetic background (SS-13^BN^ consomic rats) [17,20]. The SS-13^BN^ consomic rat, sharing 98% homology to the genome of the Dahl SS rat, exhibits normal circulating levels of ANG II on a NS diet and provides an excellent comparison for the SS rat [17].

20-HETE, a vasoconstrictor metabolite of arachidonic acid that is formed through the action of cytochrome P450-4A (CYP4A) enzymes in the vascular smooth muscle cells, has been associated with hypertension and vascular dysfunction. We recently found that a high-salt (HS; 4.0% NaCl) diet up-regulates the expression of mRNA and protein of CYP4A in the mesenteric resistance arteries of Sprague Dawley (S-D) rats [21] and cremasteric arterioles of SS rats [22]. CYP4A inhibition with dibromododecenyl methylsulfimide (DDMS) restores impaired vascular relaxation in mesenteric arteries from S-D rats fed a HS diet [21] and abolishes the potentiated vasoconstrictor response to elevated PO_2_ in cremasteric arterioles from SS rats [22]. 

Thus, it appears that two different mechanisms (an activated CYP4A/20-HETE system and chronically suppressed plasma ANG II levels) can both explain the impaired vascular relaxation in NS-fed SS rats and other strains fed a HS diet. The question is whether these two systems are working in parallel to one another or whether they are in some way interrelated. Of particular interest is the question of whether there is a relationship between circulating ANG II levels and regulation of the CYP4A/20-HETE pathway. The present study evaluated the unifying hypothesis that upregulation of CYP4A enzyme expression and 20-HETE production in cerebral resistance arteries of SS rats fed NS diets and SS-13^BN^ rats fed HS diets are due to chronic exposure to low levels of ANG II. 

## 2. Materials and Methods

### 2.1. Experimental Groups

Eight- to twelve-week-old male SS-13^BN^ consomic rats (30 animals used in total for all experiments) and Dahl SS rats (25 animals used in total for all experiments) from colonies maintained at the Medical College of Wisconsin were used for these studies. Following weaning, the SS-13^BN^ rats were maintained on a normal-salt diet (0.4% NaCl; Dyets, Inc., Bethlehem, PA, USA) or switched to a high-salt (4.0% NaCl; Dyets, Inc.) diet for 3 days to suppress plasma ANG II levels. SS rats were implanted with an osmotic minipump to deliver either isotonic saline vehicle or angiotensin II (100 ng/kg/min). This dose of ANG II has previously been shown to be sufficient to restore vascular function in SS rats, but not high enough to cause the deleterious effects of elevated ANG II on endothelium-dependent vasodilation [15,23]. Following implantation of the osmotic minipumps, the animals were placed on a high-salt diet for 3 days prior to individual experimental protocols and given water ad libitum. The Medical College of Wisconsin IACUC approved all protocols.

### 2.2. Isolated Vessel Experiments

Animals were anesthetized with an intramuscular injection containing (in mg/kg) ketamine (75.0), acepromazine (2.5), and anased (10.0). The brain was removed, and the middle cerebral artery was isolated, cannulated with micropipettes and perfused and superfused with physiological salt solution (PSS) equilibrated with a 21% O_2_, 5% CO_2_, and 74% N_2_ gas mixture, as previously described [24]. The artery was maintained at a transmural pressure of 80 mmHg and the vessel diameters were measured via video microscopy.

After the 60 min control equilibration period, vessel responses to the endothelium-dependent dilator acetylcholine (ACh; 10^−10^–10^−5^ M) and the endothelium-independent nitric oxide donor sodium nitroprusside (SNP; 10^−12^–10^−4^ M) were determined. The vessels were then incubated for 30 min in the presence of the CYP4A enzyme inhibitor DDMS (50 µM) and the responses to ACh and SNP were repeated. Another group of vessels was incubated for 30 min without the inhibitor and responses to vasodilator stimuli were measured as a time control.

In a second series of experiments, the cytochrome P450 inhibitor DDMS (50 µM) was added to the perfusate and superfusate for the final 30 min of the control equilibration period and responses of the arteries to the vasodilator agonists were recorded before and after incubation with L-NAME (100 µM) or indomethacin (1 µM) for 30 min in the perfusate and superfusate. The responses to ACh were also determined in a separate group of vessels after a 30 min equilibration period without the inhibitor as a time control.

The maximum diameter of the isolated vessel was measured at the conclusion of the experiment. Hydrogen peroxide (H_2_O_2_; 1.76 mM) was added to the superfusate to induce maximal dilation. The percent of active resting tone (%) was determined as [(D_max_ − D_rest_) / D_max_] × 100, where D_max_ is the maximum diameter in the presence of H_2_O_2_ and D_rest_ is the resting control diameter.

### 2.3. 20-HETE Production in Cerebral Arteries

Vascular 20-HETE production was measured in cerebral arteries from NS- or HS-fed SS-13^BN^ consomic rats and in arteries from ANG II- or saline-infused HS-fed SS rats as previously described [25]. In those experiments, the cerebral circulation was flushed with ice-cold Tyrode’s solution and the cerebral arteries were carefully removed and incubated in potassium phosphate buffer containing 2 mM NADPH, 40 μM cold arachidonic acid, and 2 μM indomethacin for a 1.0 mL total reaction volume at 37 °C and equilibrated with 100% O_2_ for 90 min to produce 20-HETE. The vessels were homogenized and 30 μL of the homogenate was used to measure protein concentration using a modified Bradford method. 20-HETE was extracted from the remaining homogenate using ethyl acetate, and the organic layer was dried under nitrogen. The samples were reconstituted in methanol and 20-HETE was measured using liquid chromatography-mass spectrometry (LC-MS) [25].

### 2.4. Dihydroethidium Fluorescence

Dihydroethidium (DHE) was utilized to assess the vascular reactive oxygen species (ROS) levels in basilar arteries [15]. Basilar arteries were used as a surrogate for MCA because both vessels have a NO-dependent dilation response to ACh, but the increased size of the basilar artery improves vascular imaging [15]. The basilar artery was incubated for 1 h in PSS warmed to 37 °C. DHE (5 μmol/L) was added to the incubating solution for the final 15 min. The arteries were sliced into 10 µm transverse sections and imaged with a Nikon Eclipse TS100 microscope (Nikon, Tokyo, Japan) equipped with a ×20 objective, a 540 nm excitation filter, a 605 nm emission filter (Chroma Technology Corp., Bellows Falls, VT, USA), and QImaging Regiga-2000R digital camera (Surrey, BC, Canada). Multiple images of each artery were quantified using ImageJ software and background fluorescence was subtracted from the fluorescence value of the imaged vessel [15].

### 2.5. Statistical Analysis

The data are presented as mean ± SEM. An unpaired Student’s *t*-test was used for comparisons between two groups. A one-way analysis of variance (ANOVA) was used to compare the concentration–response curves between multiple groups at each concentration. Following the ANOVA, the differences between individual groups were evaluated using a post hoc Student–Newman–Keuls test. The differences between groups were considered significant at *p* < 0.05. 

## 3. Results

### 3.1. Vascular Responses

#### 3.1.1. Vessel Diameter and Active Tone in SS-13^BN^ Consomic Rats and Dahl SS Rats Receiving Osmotic Minipumps

Maximum diameter, resting diameter, and active resting tone in isolated middle cerebral arteries (MCA) from SS-13^BN^ consomic rats and HS-fed SS rats receiving osmotic minipumps are presented in Table 1 and Table 2, respectively. The active tone in arteries from SS-13^BN^ consomic rats was unaffected by dietary salt or incubation with inhibitors (DDMS, L-NAME, indomethacin) (Table 1) and there was no effect of incubation with these inhibitors on active tone in SS rats receiving either ANG II or saline infusion (Table 2). Therefore, any effects on the magnitude of vascular relaxation responses in the present study did not result from initial differences in resting tone due to pre-existing constriction of the artery. There was a reduced resting diameter in SS + ANG II when incubated with L-NAME, likely a result of the loss of ANG II-dependent NO preservation and the vasoconstrictor effect of ANG II acting through the AT_1_ receptor, which did not change the conclusions drawn from the data collected since it did not alter the % active tone.

#### 3.1.2. Effect of Chromosome 13 Substitution on Responses of MCA to Acetylcholine (ACh)

Figure 1A summarizes the response to ACh in the MCA from SS-13^BN^ consomic rats. When the animals were fed a NS diet to maintain normal plasma ANG II levels [17], the MCA dilated in response to ACh and the addition of DDMS did not augment the dilation. By contrast, the MCA from SS-13^BN^ rats failed to respond to ACh when plasma ANG II levels were suppressed with a high-salt diet. Addition of DDMS to inhibit CYP4A enzymes fully restored ACh-induced dilation to NS-control values in the arteries of HS-fed SS-13^BN^ rats. L-NAME abolished the response to ACh in the MCA from NS-fed SS-13^BN^ and in DDMS-treated MCA from high-salt-fed SS-13^BN^ consomic rats (Figure 1B), showing that vascular relaxation to ACh in both those groups is nitric oxide-dependent.

The MCA from SS-13^BN^ consomic rats fed either a NS or HS diet dilated normally to the exogenous nitric oxide donor, sodium nitroprusside. Addition of DDMS to the tissue bath did not affect this dilation (Figure 1C), indicating that the vascular dysfunction in SS-13^BN^ rats fed a HS diet is due to a loss of nitric oxide bioavailability, rather than a failure of the vascular smooth muscle cells to respond to NO.

#### 3.1.3. Effect of Low-Dose ANG II Infusion on Vascular Responses in MCA from Dahl SS Rats

Figure 2A summarizes the response to ACh in MCA from ANG II-infused SS rats versus saline-infused SS rats fed a HS diet. When circulating ANG II levels were elevated by delivering a sub-pressor dose of ANG II via an osmotic minipump, ACh caused vascular relaxation with no additional benefit of acute treatment with DDMS. Saline-treated SS rats failed to respond to ACh, but the addition of DDMS to inhibit CYP4A restored ACh-induced vasodilation in arteries from those animals. Similar to our findings in the SS-13^BN^ consomic rats, these data indicate that maintenance of normal circulating levels of ANG II leads to improved vascular relaxation. 

The vascular response to ACh in the MCA from both ANG II-infused SS rats and DDMS-treated MCA from saline-treated SS rats utilizes nitric oxide, as ACh-induced dilation in both groups was eliminated by co-incubation with L-NAME and was unaffected by indomethacin in the tissue bath (Figure 2B). The MCA from SS rats receiving either ANG II or saline ± DDMS responded normally to sodium nitroprusside (Figure 2C), demonstrating that the vascular dysfunction in the SS rat is due to loss of NO bioavailability, rather than reduced responsiveness of the vascular smooth muscle cells to NO per se.

### 3.2. Vascular 20-HETE Production

#### 3.2.1. 20-HETE Levels in Cerebral Arteries of SS-13^BN^ Rats

The production of 20-HETE by CYP4A enzymes (Figure 3A) in cerebral arteries from SS-13^BN^ consomic rats were unaffected by dietary salt content. These data do not support the hypothesis that ANG II suppression is causing an increase in 20-HETE production in the SS rat by CYP450 enzymes. As such, these findings are contrary to the vascular data presented in Figure 1 demonstrating that CYP4A inhibition restores vascular relaxation in response to acetylcholine in HS-fed SS-13^BN^ consomic rats. Because elevated dietary salt intake does not affect vascular 20-HETE production, a high-salt diet must be indirectly unveiling a role for the CYP4A/20-HETE pathway in vascular dysfunction in SS-13^BN^ consomic rats. A likely candidate for this indirect effect is the HS-induced elevation of tissue reactive oxygen species. 

#### 3.2.2. 20-HETE Levels in Cerebral Arteries from Saline- or ANGII-Infused Dahl SS Rats

Figure 3B compares 20-HETE production in cerebral arteries from SS rats receiving either saline or a low-dose ANG II infusion. ANG II infusion had no effect on 20-HETE production in the cerebral arteries of HS-fed SS rats. Similar to our findings in SS-13^BN^ consomic rats, these data indicate that CYP4A regulation is independent of changes in circulating ANG II levels, because neither salt-induced ANG II suppression in the SS-13^BN^ consomic rat nor artificial ANG II supplementation in the low-renin SS rat altered cerebral vascular 20-HETE production.

### 3.3. Dihydroethidium (DHE) Fluorescence

#### Vascular Reactive Oxygen Species Evaluated by DHE Fluorescence

Figure 4 summarizes the levels of reactive oxygen species (ROS) in arteries from SS-13^BN^ consomic rats fed a NS or HS diet with and without inhibition of CYP4A enzymes with DDMS. Restoring circulating ANG II levels by introgression of Brown Norway chromosome 13 (SS-13^BN^ consomic rat) reduced vascular ROS to the same extent as incubating vessels from HS-fed SS-13^BN^ with the CYP4A inhibitor DDMS. These data demonstrate a common mechanism for the reduction of vascular ROS levels between these two independent pathways (restoration of normal ANG II levels and inhibition of the CYP4A pathway), ultimately resulting in the restoration of vascular function.

## 4. Discussion

The present study sought to evaluate the possible contribution of the CYP4A/20-HETE pathway to the vascular dysfunction in the SS rat. In Sprague Dawley (S-D) rats, high-salt-induced suppression of the renin–angiotensin system (RAS) is accompanied by a significant elevation of CYP4A protein expression in mesenteric arteries [21] suggesting that the RAS is a potential pathway to exert regulatory control over cytochrome P450-4A (CYP4A). Based on those initial studies, we hypothesized that there is a negative feedback loop whereby the renin–angiotensin system exerts a chronic inhibitory effect over CYP4A/20-HETE expression. Consistent with this hypothesis, the elevated CYP4A protein observed in mesenteric arteries from S-D rats fed high-salt diet has been shown to contribute to vascular dysfunction [21], possibly as an indirect effect of suppressed plasma ANG II.

The existence of a consomic panel of SS rats with individual chromosomes from the normotensive Brown Norway (BN) rat introgressed into the Dahl SS genetic background [20] provides a unique opportunity to evaluate the interrelationship between the renin–angiotensin system and the CYP4A/20-HETE pathway in the SS rat. The genetic homogeneity of the consomic animal model has reduced phenotypic noise, making it easier to detect the possible contribution of individual chromosomes to functional differences in complex, multifactorial diseases such as hypertension. The SS-13^BN^ consomic rat has chromosome 13 from the BN rat introgressed on the SS genetic background. This strain is more than 98% identical to the SS rat genetically [20], but has a normally functioning renin–angiotensin system, resulting in physiologic levels of plasma ANG II when the animals are fed a normal-salt diet and suppression of plasma ANG II following ingestion of elevated dietary sodium [20]. 

In the present study, the vascular responses to ACh were intact in SS-13^BN^ consomic rats fed a NS diet. In agreement with previously published reports [26,27], this is presumably due to restoration of the normal function of the renin–angiotensin system, which would produce physiological levels of circulating ANG II. In the present study, endothelium-dependent vascular relaxation in response to ACh was abolished when the SS-13^BN^ consomic rats were fed a high-salt diet for three days to suppress the renin–angiotensin system. Previously published reports on the effect of chronic exposure to low levels of ANG II on antioxidant capabilities, as observed in HS-fed S-D rats [28] and NS-fed SS rats [15], describe reduced expression of Cu/Zn-SOD (and possibly other antioxidant enzymes), resulting in reduced antioxidant defenses. In addition to the reduction in antioxidant capabilities, it is possible that a pro-oxidant pathway capable of increasing ROS production is also unveiled under these conditions.

In the present study, treating MCA from high-salt-fed SS-13^BN^ consomic rats with the CYP4A inhibitor DDMS to reduce 20-HETE production fully restored the response to ACh to NS-fed control values, similar to the effects of restoring normal plasma ANG II levels. Taken together, these findings suggest that either two independent mechanisms lead to impaired vascular relaxation in HS-fed SS-13^BN^ rats or that there is a potential interaction between these two pathways. One possibility is a role of circulating ANG II in the regulation of the CYP4A/20-HETE pathway by stimulating CYP4A enzyme activity to increase 20-HETE production. If this direct connection does not exist, there must be a separate mechanism for the CYP4A/20-HETE pathway to play a role in the vascular dysfunction with high-salt diet or in an ANG II-suppressed state. 

Even though the CYP4A/20-HETE pathway appears to be contributing to vascular dysfunction in high-salt-fed SS-13^BN^ consomic rats, there was no significant difference in 20-HETE production in the cerebral arteries of NS- and HS-fed SS-13^BN^ rats in the present study. It is possible to affect the CYP4A/20-HETE pathway by altering the expression of individual CYP4A isoforms, having different catalytic efficiencies to form 20-HETE. The measurement of 20-HETE production in cerebral resistance arteries, however, argues against this possibility because there was no significant difference in the amount of 20-HETE produced in cerebral arteries from SS-13^BN^ consomic rats fed either a normal-salt or a high-salt diet. As such, these data do not support a role for salt-induced ANG II suppression in directly regulating the CYP4A/20-HETE pathway.

As mentioned previously, the use of consomic animals has limitations due to the genetic material carried on chromosome 13 in addition to the renin allele. Thus, it is possible that other genes or gene regulators accompanying the renin gene on chromosome 13 exert a functional impact on the CYP4A/20-HETE pathway. In order to confirm our findings, we evaluated the direct effect of plasma ANG II on the CYP4A/20-HETE system by repeating our vascular studies in MCA from ANG II-infused SS rats. In these studies, supplementing circulating ANG II in the low-renin SS rat restored vascular relaxation responses, providing another example of the pervasive deleterious effect of suppressed ANG II on vascular function. The MCA from saline-treated SS rats failed to respond to ACh in control conditions. However, inhibition of CYP4A enzyme activity fully restored endothelium-dependent vasodilation in response to ACh in those animals. From those data, it appears that suppressed plasma ANG II in saline-treated SS rats (similar to high-salt-fed SS-13^BN^ consomic rats), unveils a CYP4A/20-HETE effect on vascular function that is not observed when normal physiological levels of angiotensin II are maintained via an osmotic minipump. 

20-HETE production in cerebral arteries from SS rats with minipump implantation showed no effect of ANG II on the CYP4A/20-HETE pathway itself. Considering the consistency between the ANG II-infused SS rat data and our findings in the SS-13^BN^ consomic rat, it is reasonable to conclude that alterations in circulating ANG II levels do not have a direct regulatory effect on the CYP4A/20-HETE pathway in the SS rat, but instead both pathways have independent influences on vascular function. Thus, the effects of salt on the CYP4A/20-HETE pathway previously observed in S-D rats [21] seem to be either altered or nonexistent in the SS rat. 

Vascular reactive oxygen species significantly contribute to vascular dysfunction in the SS rat [18,19]. The MCA from SS rats treated chronically with the superoxide dismutase mimetic tempol in the drinking water showed healthy vascular relaxation responses to acetylcholine and reduced PO_2_ regardless of dietary salt intake [18,19]. These findings indicate that ROS contribute to endothelial dysfunction in the SS rat fed either a normal-salt diet or high-salt diet. In the S-D rat, a high-salt diet stimulates ROS production and high-salt-induced ANG II suppression results in lowered Cu/Zn-SOD expression, thereby compromising antioxidant defense mechanisms and further elevating ROS levels [10,28]. CYP450-4A enzymes also have the capacity to produce superoxide anions (O_2_^−^) through their normal catalytic activity [29,30], via stimulation of NADPH oxidase [31,32,33,34], and by directly uncoupling endothelial nitric oxide synthase (eNOS), resulting in production of O_2_^−^ instead of nitric oxide [35,36]. 

For these reasons, it is likely that the common factor between the HS-fed S-D rat and the low-renin SS rat is an elevation of ROS and not, as we had predicted, a direct effect of the renin–angiotensin system on the CYP4A/20-HETE pathway. Indeed, direct evaluation of ROS in basilar arteries from both NS-fed SS-13^BN^ consomic rats and ANG II-infused Dahl SS rats showed significantly lower levels of vascular ROS than in their counterparts with suppressed plasma ANG II levels. Furthermore, incubating arteries from HS-fed SS-13^BN^ consomic rats and saline-infused Dahl SS rats with the CYP4A inhibitor DDMS significantly reduced vascular ROS, to the same extent as restoring normal circulating ANG II levels, suggesting that both pathways influence vascular oxidative stress. Taken together, the current findings demonstrate an interesting model whereby two independent pathways affect vascular function similarly through a common mechanism. 

## 5. Conclusions

Previous studies in our laboratory have shown that both the renin–angiotensin system and the CYP4A/20-HETE pathway play a role in the vascular dysfunction in the Dahl salt-sensitive rat. This study demonstrated that normalization of circulating ANG II levels or inhibition the CYP4A/20-HETE pathway restores vascular function in this animal model. The novel finding from this research is that these two pathways appear to influence vascular function independent of each other while sharing the common mechanism of generating vascular oxidative stress. 

## Figures and Tables

**Figure 1 antioxidants-12-00783-f001:**
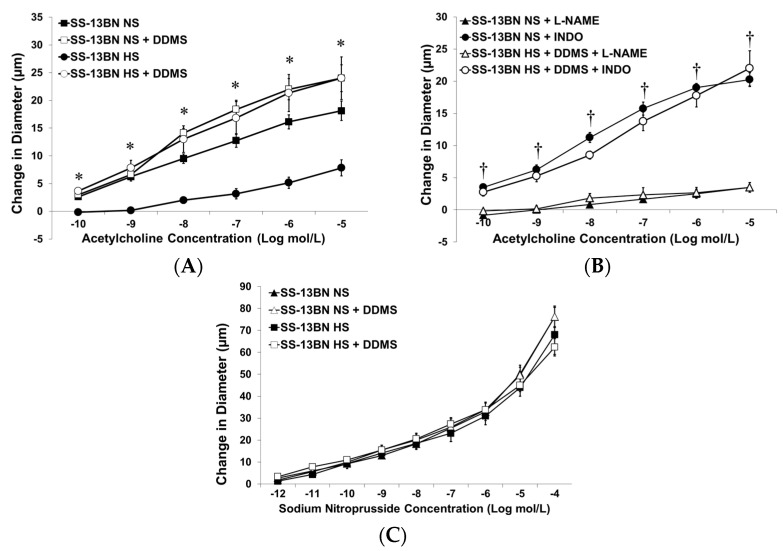
Response of isolated middle cerebral arteries of SS-13^BN^ consomic rats fed normal-salt (NS) diet or high-salt (HS) diet to: (**A**) acetylcholine (ACh; 10^−10^–10^−5^ M) ± acute addition of 50 µM DDMS; (**B**) ACh ± DDMS (50 µM) and ±L-NAME (100 µM) or indomethacin (INDO; 1 µM); and (**C**), sodium 50 µM nitroprusside (10^−12^–10^−4^ M) ± acute addition of DDMS (50 µM) to the tissue bath. Data are expressed as mean change in diameter (µm) from baseline ± SEM and n = 4–8 for all groups. Significant difference: * *p* < 0.05 vs. arteries from HS-fed SS-13^BN^ consomic rats, † *p* < 0.05 vs. L-NAME-treated vessels.

**Figure 2 antioxidants-12-00783-f002:**
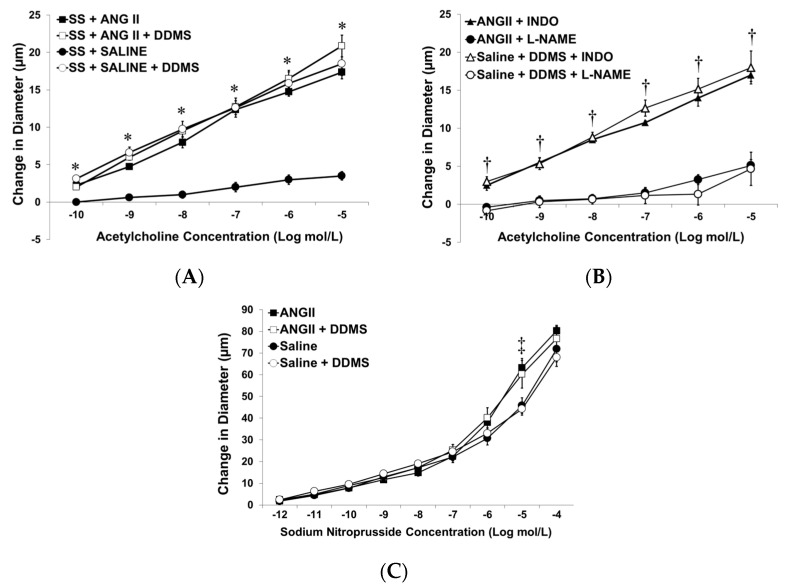
Response of isolated middle cerebral arteries of SS rats fed high-salt diet and receiving infusion of either isotonic saline or angiotensin II (ANG II; 100 ng/kg/min) from an osmotic minipump to: (**A**) acetylcholine (ACh; 10^−10^–10^−5^ M) ± acute addition of DDMS (50 µM); (**B**); ACh ± DDMS (50 µM) and/or ±L-NAME (100 µM) or indomethacin (INDO; 1 µM); and (**C**) sodium nitroprusside (10^−12^–10^−4^ M) ± acute addition of DDMS (50 µM) to the tissue bath. Data are expressed as mean change in diameter (µm) from baseline ± SEM, and n = 6–8 for all groups. Significant difference: * *p* < 0.05 vs. saline-infused SS rats, † *p* < 0.05 vs. L-NAME-treated vessels, ‡ *p* < 0.05 vs. saline-infused SS ± DDMS at the 10^−5^ M dose.

**Figure 3 antioxidants-12-00783-f003:**
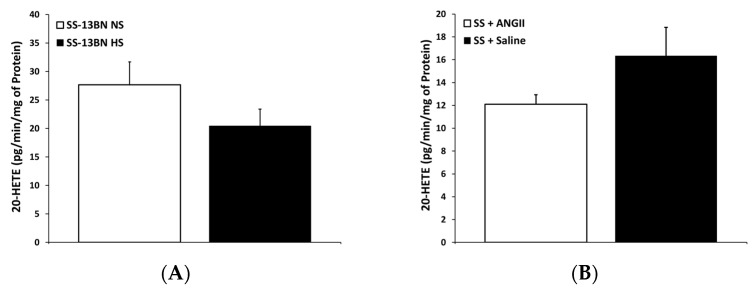
20-HETE production in cerebral arteries from (**A**) normal-salt (NS)- and high-salt (HS)-fed SS-13^BN^ consomic rats and (**B**) SS rats fed high-salt diet with minipump delivery of either isotonic saline or angiotensin II (ANG II; 100 ng/kg/min). 20-HETE production assessed via LC-MS (n = 4–5 for all groups). Data are presented as amount of 20-HETE (pg) produced per minute per mg of protein.

**Figure 4 antioxidants-12-00783-f004:**
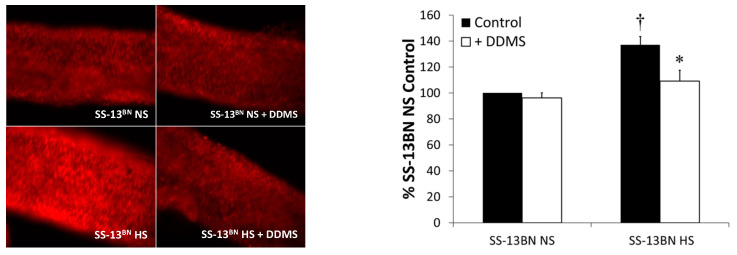
Vascular reactive oxygen species evaluated by DHE fluorescence in basilar arteries of normal-salt (NS)- and high-salt (HS)-fed SS-13^BN^ consomic rats in either control conditions or incubated with DDMS (n = 4–5 for all groups). Data are presented as % of DHE fluorescence (raw fluorescence units) in basilar arteries from NS-fed SS-13^BN^ rats. * *p* < 0.05, significantly different from HS-fed SS-13^BN^; † *p* < 0.05, significantly different from NS-fed SS-13^BN^ ± DDMS.

**Table 1 antioxidants-12-00783-t001:** MCA Diameters and Resting Tone in SS-13^BN^ Consomic Rats.

Experimental Groups	Number (n)	Max Diameter (µm)	Rest Diameter (µm)	Active Tone (%)
SS-13^BN 1^ NS ^2^	6	236 ± 2.0	137 ± 4.7	42 ± 2.1
SS-13^BN^ HS ^3^	6	245 ± 3.9	149 ± 3.3	39 ± 2.0
SS-13^BN^ NS + DDMS ^4^	6	237 ± 2.0	140 ± 6.0	40 ± 4.7
SS-13^BN^ HS + DDMS	6	245 ± 3.9	133 ± 5.16	45 ± 2.9
SS-13^BN^ NS + L-NAME ^5^	6	246 ± 5.9	120 ± 10.1	51 ± 3.6
SS-13^BN^ HS + DDMS + L-NAME	6	247 ± 4.7	134 ± 6.3	45 ± 2.3
SS-13^BN^ NS + INDO ^6^	4	257 ± 8.5	147 ± 14.6	45 ± 7.5
SS-13^BN^ HS + DDMS + INDO	4	239 ± 3.6	150 ± 7.3	37 ± 3.4

Data are summarized as mean ± SEM. ^1^ SS-13^BN^: SS-Chr 13^BN^/Mcw, ^2^ NS: Normal-salt (0.4% NaCl), ^3^ HS: High-salt (4% NaCl), ^4^ DDMS: N-methylsulfonyl-12,12-dibromododec-11-enamide, ^5^ L-NAME: N^ω^-nitro-L-arginine methyl ester, ^6^ INDO: Indomethacin.

**Table 2 antioxidants-12-00783-t002:** MCA Diameters and Resting Tone in Dahl SS Rats with Minipump.

Experimental Groups	Number (n)	Max Diameter (µm)	Rest Diameter (µm)	Active Tone (%)
SS ^1^ + ANG II ^2^	8	231 ± 2.6	135 ± 2.0	42 ± 1.3
SS + Saline	8	235 ± 3.4	141 ± 3.2	40 ± 1.8
SS + ANG II + DDMS ^3^	8	231 ± 2.6	145 ± 4.9	37 ± 2.5
SS + Saline + DDMS	8	235 ± 3.4	142 ± 5.2	40 ± 2.7
SS + ANG II + L-NAME ^4^	6	240 ± 3.9	115 ± 5.4 *	52 ± 2.4
SS + Saline + DDMS + L-NAME	6	240 ± 4.0	125 ± 5.0	48 ± 2.0
SS + ANG II + INDO ^5^	3	248 ± 8.7	151 ± 6.3	37 ± 1.7
SS + Saline + DDMS + INDO	3	236 ± 3.5	138 ± 5.0	41 ± 2.3

Data are summarized as mean ± SEM. ^1^ SS: Dahl SS, ^2^ ANG II: angiotensin II, ^3^ DDMS: N-methylsulfonyl-12,12-dibromododec-11-enamide, ^4^ L-NAME: N^ω^-nitro-L-arginine methyl ester, ^5^ INDO: indomethacin. * *p* < 0.05, significantly different from every treatment group within the same category.

## Data Availability

The data that support the findings of this study are available from the corresponding author, J.L., upon reasonable request.

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
