# Peer review of "Effect of Chronically Suppressed Plasma Angiotensin II on Regulation of the CYP4A/20-HETE Pathway in the Dahl Salt-Sensitive Rat"

_antioxidants, 2023, doi:10.3390/antiox12040783_

Round 1
Reviewer 1 Report
The manuscript "Effect of Chronically Suppressed Plasma Angiotensin II on Regulation of the CYP4A/20-HETE Pathway in the Dahl Salt-Sensitive Rat" discusses the role of the renin-angiotensin system and the CYP4A/20-HETE pathway in vascular dysfunction in Dahl salt-sensitive (SS) rats. The study uses a consomic panel of SS rats with individual chromosomes from the normotensive Brown Norway (BN) rat to evaluate the interrelationship between the renin-angiotensin system and the CYP4A/20-HETE pathway. The study finds that both pathways play a direct role in vascular dysfunction in SS rats but are independent of each other, even though they may both contribute to vascular dysfunction through ROS production. The study provides a unique opportunity to evaluate the interrelationship between two pathways using the consomic animal model, which has reduced phenotypic noise, making it easier to detect the possible contribution of individual chromosomes to functional differences in complex, multifactorial diseases such as hypertension. The findings suggest that either two independent mechanisms lead to impaired vascular relaxation in high salt-fed SS-13BN rats or that there is a potential interaction between these two pathways. However, further studies are needed to investigate the potential interaction between the renin-angiotensin system and the CYP4A/20-HETE pathway.
Better quality of graphics would be desirable
Overall, the manuscript provides good insights into the pathogenesis of vascular dysfunction in Dahl SS rats and potential therapeutic targets for hypertension.
Author Response
Response to reviewer 1 attached as a Word document. Please see attachment.

Reviewer 2 Report
The manuscript by Lukaszewicz et al. entitled: "Effect of Chronically Suppressed Plasma Angiotensin II on Regulation of the CYP4A/20-HETE Pathway in the Dahl Salt-Sensitive Rat" is designed to answer the question of whether there is a relationship between circulating levels of ANG II and regulation of the CYP4A/20-HETE pathway. To achieve their goal, the authors performed experimental protocols in SS rats fed a normal diet and SS-13BN rats fed a high-salt diet, as both animal models have chronically low plasma levels of ANG II. The experiments were performed in resistance cerebral arteries and in the basilar artery. Their results demonstrate that the renin-angiotensin system and the CYP4A/20-HETE pathway play a direct role in vascular dysfunction, but these mechanisms are independent of each other, although both may contribute to vascular dysfunction through ROS production. Overall, this is a well-designed study and the topic is interesting.
I have some recommendations:
The authors should define MCA in the abstract (line 22). They indicated that they used basilar arteries (line 19), then in the same abstract they indicated MCA without defining this acronym. This can be confusing.
Authors should consider including that they study cerebral circulation in the title and/or keywords.
Be consistent when writing CYP4A (with or without hyphen). For example, line 93 CYP-4A.
The basilar artery is only used to measure ROS levels. The authors explain why they used the basilar artery instead of the MCA in lines 120-122. This reviewer understands how difficult it is to work with these vessels, and the explanation that both vessels have NO-dependent dilation to Ach. However, why did the authors not perform isolated vascular experiments or measurement of 20-HETE in the basilar artery? The local mechanism may vary from vessel to vessel.
Although the number of animals used in each experiment is indicated in each result (tables and figures), the authors should indicate the total number of animals per group used in this study in the materials and methods section.
Table 2 shows significant differences with L-NAME incubation in the SS+ANGII group at rest diameter, could this result be explained?
Ang II has a vasocontractile effect through activation of ATR1 in the vascular smooth muscle cell. In addition, Ang II-induced contraction has also been linked to superoxide anion production at the vascular level. However, Lukaszewicz et al. explain in their article that chronic low levels of Ang II can cause an increase in ROS and consequent endothelial dysfunction. Please can you explain this mechanism more precisely in the introduction, and is it related to AT2 receptors?
The authors also indicate that there is a narrow window of Ang II doses that were sufficient to restore vascular function but not enough to cause deleterious effects of elevated Ang II. Could you expand on this idea in the discussion by indicating Ang II concentrations?
Perhaps citation 3 is inappropriate, as the words renin and angiotensin do not appear in the article.
Author Response
Response to reviewer 2 has been uploaded as a Word document. Please see the attachment.
